# Changes in Protein Non-Covalent Bonds and Aggregate Size during Dough Formation

**DOI:** 10.3390/foods9111643

**Published:** 2020-11-11

**Authors:** Sonoo Iwaki, Shiro Aono, Katsuyuki Hayakawa, Bin Xiao Fu, Chikako Otobe

**Affiliations:** 1Cereal Science Research Center of Tsukuba, Nisshin Flour Milling Inc., 13 Ohkubo Tsukuba, Ibaraki 300-2611, Japan; iwaki.sonoo@nisshin.com (S.I.); aono.shiro@nisshin.com (S.A.); 2Degree Programs in Life and Earth Science, Graduate School of Science and Technology, University of Tsukuba, 1-1-1 Tennodai Tsukuba, Ibaraki 305-8577, Japan; ochika@affrc.go.jp; 3Grain Research Laboratory, Canadian Grain Commission, 303 Main Street, Winnipeg, MB R3C 3G8, Canada; binxiao.fu@grainscanada.gc.ca; 4Institute of Crop Science, National Agriculture and Food Research Organization, 2-1-2 Kannondai Tsukuba, Ibaraki 305-8518, Japan

**Keywords:** wheat, flour, dough, mixing, non-covalent bond, aggregation

## Abstract

This research investigated changes in the amounts and sizes of monomeric proteins and protein aggregates during dough mixing, with a focus on the contribution of non-covalent bonds in the aggregation of gluten proteins. High protein flour (HF) and low protein flour (LF) were used in this study. As dough mixing progressed from flour to overmixed dough, the total amount of protein aggregates increased while the amount of monomeric protein decreased. Omega-gliadin was the major monomeric protein that decreased in quantity. Interestingly, the amount of larger-sized protein aggregates decreased and that of smaller-sized protein aggregates increased. The amount of gluten protein macro-polymer aggregated through strong non-covalent bonds decreased whereas aggregates formed with weaker non-covalent bonds increased. LF dough behaved similar to HF dough. Large-sized gluten protein aggregates disaggregated due to the weakening of non-covalent bonds and became smaller. Omega-gliadin was incorporated into gluten protein aggregates during dough mixing.

## 1. Introduction

Mixing is an important process in bread making. When water and wheat flour are mixed with sufficient energy input, proteins in the wheat flour are hydrated and gluten networks are formed in the dough. Many schematic models have been proposed to describe dough structure [1,2,3,4] and the dough development process [5,6]. MacRitchie [7,8], Belton [9], and Vliet and Hamer [10] proposed conflicting ideas regarding the structural organization of the protein polymers in dough formation and this debate continues to this day [11]. 

Disulfide bonds and non-covalent bonds are both important for gluten functionality and dough structure. The amino acid composition of wheat gluten proteins is unique, with glutamine and proline comprising ~35% and ~10% of the total amino acids, respectively [12]. Glutamine facilitates the formation of hydrogen bonds and proline provides hydrophobic interactions. Non-covalent bonds are affected by the presence of salt. Sodium chloride shields the charges on gluten molecules and weakens electrostatic repulsion between protein molecules, resulting in stronger dough [13,14].

Belton [15] proposed a loop and train model for gluten proteins. This model focuses on the role of hydrogen bonds in protein structure and suggests two distinct regions in the glutenin molecule: one that is stabilized by interchain hydrogen bonds (train) and another that is un-bonded (loop). Hydration is believed to reduce the size of the train region with a β-sheet conformation and increase the size of the hydrated loop region, as indicated by the formation of extended hydrated β-turn structures. Studies of gluten proteins using Fourier-transform infrared spectrometry (FT-IR) have shown that the amount of β-sheet structure increases during dough mixing [16,17,18].

The front-face fluorescence method [19] was recently used to investigate changes in hydrophobic interactions between protein molecules during dough mixing, based on the degree of hydrophobicity of the dough surface [18]. The authors suggested that hydrophobic interactions between proteins become stronger during dough mixing because of the decrease in the surface hydrophobicity of the dough.

The role of non-covalent bonds in dough formation is only partially understood due to the difficulty of analyzing non-covalent bonds. The objective of this study was to investigate the changes in the molecular size of gluten proteins and changes in the non-covalent bonds between proteins in dough during mixing to better understand the mechanism underlying dough formation. Sodium dodecyl sulfate (SDS) reduces the amount of hydrophobic interactions within protein polymers [20], and provides electrical charges. Thus, SDS aqueous solution weakens non-covalent bonds such as hydrogen, ionic, and hydrophobic interactions. In this study, we analyzed the size distribution of proteins extracted with various concentrations of SDS and examined the relationship between non-covalent bonds and the size of protein aggregates. The amount of protein aggregates that can be extracted from the dough depends on the SDS concentration. As the concentration of SDS increases, the degree of non-covalent bond disruption increases. Proteins aggregated through weak non-covalent bonds can be extracted using a low concentration (0.1%, in this study) of SDS. The difference between the amount of protein extracted with medium (0.3%) and low concentrations of SDS is due to macro-polymers aggregated through medium-strength non-covalent bonds. The amount of gluten protein aggregated through strong non-covalent bonds is calculated by the difference in the amount of protein extracted with high (0.5%) and medium concentrations of SDS. To extract all proteins in flour or dough, we used SDS coupled with sonication, as described by Singh et al. [21]. Size-exclusion high-performance liquid chromatography (SE-HPLC) is widely used to analyze protein size distribution but is of limited value for very large protein aggregates and molecules because of its exclusion limit [22] and thus here we used field flow fractionation (FFF) technology to analyze very large protein aggregates.

## 2. Materials and Methods

### 2.1. Wheat Flour

We used two commercial flour samples (Nisshin Flour Milling Inc., Tokyo, Japan). One was milled from hard wheat and had a protein content of 14.5% (dry basis) and the other was milled from semi-hard wheat and had a protein content of 11.7%. These flour samples are referred to as high protein flour (HF) and low protein flour (LF).

### 2.2. Dough Sampling during Mixing

Flour (200 g), distilled water (HF: 128.8 mL, LF: 117.4 mL; mean water absorption for farinograph), and salt (4 g) were mixed in a Swanson mixer (National Mfg. Co., Lincoln, NE, USA) at a constant temperature of 27 °C. Each dough was mixed at 120 rpm for 20 min to achieve overmixing conditions. The change in electric power during mixing was monitored with an AF-1700 data logger (ATTO Co., Tokyo, Japan). The mixing curves are shown in Figure 1. Dough samples were collected at the point of dough build up (HF: 4 min, LF: 6 min), peak consistency (HF: 8 min, LF: 11 min), dough breakdown (HF: 12 min, LF: 16 min), and overmixed dough (20 min). BU, PC, BD, and OM were estimated by the pattern of mixing curve.

### 2.3. Protein Extraction

Immediately after sampling, the dough sample (1 g) was homogenized with 20 mL of 50 mM phosphate buffer (pH 7.0) containing 0.5%, 0.3% or 0.1% (*w*/*v*) SDS (hereafter called ‘extraction buffer’) at 10,000 rpm for 5 min using an Ace-AM10 homogenizer (Nihonseiki Kaisha Ltd., Tokyo, Japan). Next, ultrasonic extraction was conducted using 0.5% SDS by sonication for 15 s at 60% power (76 W) using a VCX-130 sonicator (Sonics and Materials Inc., New town, CT, USA). The supernatant was collected after centrifugation at 5000× *g* for 10 min and extraction buffer (1 mL) was added to the precipitated residue. After stirring, the suspension was re-centrifuged and the supernatant collected. The supernatants were combined and mixed, and the amount of protein (N × 5.7) was determined using a Kjeltech 8400 automated protein analyzer (FOSS, Hillerod, Denmark). Each extraction for SE-HPLC and FFF analysis was repeated 6 times.

### 2.4. Analysis of Protein Size Distribution by SE-HPLC

The extracted protein solutions were brought to 100 mL with extraction buffer, then diluted twice with extraction buffer and filtered through a 0.45 μm filter. The molecular weight distribution of the proteins was analyzed by size-exclusion high-performance liquid chromatography using an ultraviolet (UV) multi-angle laser-light scattering (SE-HPLC-UV-MALLS) system comprising a 1260 Infinity HPLC system (Agilent Technologies, Santa Clara, CA, USA) consisting of an online degasser, isocratic pump, auto injector, column oven, and UV detector. Light scattering was measured (laser wavelength = 632.8 nm) using a Wyatt multi-angle laser light-scattering detector (model Dawn 8^+^, Wyatt Technology Corp., Santa Barbara, CA, USA). A Yarra 3 μm SEC-4000 size-exclusion column (300 × 7.8 mm i.d., 3 μm) (Phenomenex, Torrance, CA, USA) was used for separation. The samples (50 μL) extracted with various SDS solutions were eluted at 35 °C at a flow rate of 0.5 mL/min. 

The UV signal (214 nm) was integrated using Open LAB software (Agilent Technologies). The amount of protein in each fraction was calculated by multiplying the amount of extracted protein by the area ratio in the HPLC chromatogram.

The molecular weights (*M*_w_) of the proteins were calculated using ASTRA 6.1.6 software (Wyatt Technology Corp) employing the Berry extrapolation technique (second-order). The value of the reflective index increment (dn/dc, the change in reflective index ‘n’ with density ‘c’) used to calculate *M*_w_ was analyzed using a reflective index detector (Optilab, Wyatt Technology Corp.) and was 0.25 for 0.5% SDS extraction buffer. The UV absorption coefficient (15.3 mL/(mg × cm)) of the extracted protein was calculated by dividing the absorbance at 214 nm of the extract by the protein mass of the extract. 

### 2.5. Analysis of Protein Size Distribution by FFF

The extracted protein solutions were filtered through a 1.2 μm filter, then 20 μL aliquots were separated using an AF4 Eclipse separation system (Wyatt Technology Corp, Santa Barbara, CA, USA.) combined a 1260 Infinity HPLC system. Extraction buffer was used as the mobile phase. The detector flow rate was 1.0 mL/min and the injection flow rate was 0.5 mL/min. The cross flow rate was held constant at 2.0 mL/min for 15 min, was decreased to 0 mL/min over 10 min, remained at 0 mL/min for 10 min, then was increased to 2 mL/min and kept constant for 10 min. UV and MALLS signals were analyzed in the same manner as for SE-HPLC. 

### 2.6. Identification of Protein by 2Dimensional Fluorescence Difference Gel Electrophoresis (2D-DIGE)

Mixtures of protein aggregates (A + B) or monomeric protein (C) (Figure 2a) were collected twice and concentrated to 200 μL using a 10 kDa or 3 kDa cut-off ultrafiltration column (Vivaspin 20, Sartorius Lab Instruments, Goettingen, Germany). Next, 1.8 mL of cold acetone containing 10% trichloroacetic acid was added to the concentrated protein solution and the sample was stored at −20 °C for 4 h. After centrifugation at 14,000× *g* for 8 min, the supernatant was discarded, 1 mL of cold acetone was added, and the sample was stored at −20 °C for 10 min. After centrifugation at 14,000× *g* for 8 min, the supernatant was discarded and the residue was air-dried for 5 min. Swelling solution (15 μL) from a glycine system reagent set for Auto2D (Merck, Tokyo, Japan) was added to the residue and vortexed for 10 min. Next, 1 μL of 200 μmol IC3-Osu fluorescent dye (Dojindo Laboratories, Kumamoto, Japan) or 1 μL of 200 pmol IC5-Osu fluorescent dye (Dojindo Laboratories) dissolved in N,N-dimethylformamide and 0.5 μL of 1.5 M Tris-HCl buffer (pH8.8) was added to 10 μL of each sample at each mixing time. As an internal standard, 2 μL of sample at each mixing time was mixed with 1 μL of 200 pmol cyDye DIGE Fluor Cy2 minimal dye (Cytiva, Tokyo, Japan) and 0.5 μL of 1.5 M Tris-HCl buffer (pH 8.8). The dye-containing internal standard and each sample was stored at 4 °C in the dark for 1 h for labeling. Next, 1 μL of 10 mM lysine was added to the labeled internal standard and each sample and the mixtures were stored at 4 °C for 10 min. Internal standard (2 μL) labeled with Cy2, 2 μL of sample labeled with IC3, 2 μL of sample labeled with IC5, and 7 μL of working swelling fluid (swelling fluid: 1 M dithiothreitol: ampholyte =113.4:6:0.6) were mixed. Each prepared mixture was individually loaded onto the 2D-DIGE and analysis was performed using Auto-2D Plus (Merck). The electrophoresis conditions for the desalting mode were as recommended by the manufacturer using pH 3–10 isoelectric focusing (IEF) chips and 10% polyacrylamide gel electrophoresis (PAGE) chips (Merck). After electrophoresis, the gels were removed from the plate, photographed with an Amersham Typhoon imager (Cytiva), and Ettan DIGE analysis was performed using Melanie 9 software (Cytiva).

### 2.7. Statistics

Significant differences (*p* < 0.05, *n* = 6) for SE-HPLC analysis were determined by Tukey’s method using JMP software (SAS Institute Inc., Cary, NC, USA). Significant differences (*p* < 0.05, *n* = 4) for 2D-DIGE analysis were determined by the analysis of variance (ANOVA) test using Melanie 9 software (Cytiva).

## 3. Results 

### 3.1. Changes in the Amount of Extracted Protein during Mixing

The amount of protein extracted from HF or LF with 0.5% SDS buffer is shown in Figure 3. In a preliminary experiment, we used 0.5% SDS buffer so that the amount of protein extracted with buffer containing more than 0.5% SDS did not differ from the amount of protein extracted with 0.5% SDS buffer. The extraction efficiencies were 91–93%. The amount of protein extracted from HF was higher than that LF because the original protein content of HF is higher than that of LF. There was no significant difference in protein solubility at each stage of mixing. Protein extraction efficiencies increase during mixing [23,24,25]. Singh et al. showed that the extraction efficiency of protein by ultrasonication is 92.8%, and this level of extraction is considered “complete extraction” [21]. We also extracted proteins using ultrasonication and achieved “complete extraction”, showing that extraction efficiency did not increase with increased dough mixing. 

### 3.2. Change in Protein Size Distribution during Mixing 

A SE-HPLC-UV chromatogram and a molecular weight curve of proteins extracted from HF using 0.5% SDS buffer are shown in Figure 2a. The molecular weight curve has significantly different slopes at points ‘a’ and ‘b’. Point ‘b’ is the cut-off point separating monomeric protein and protein aggregates. Proteins were fractioned according to their elution times. The protein fractions ‘a’, ‘b’, and ‘c’ were defined as protein A aggregates, protein B aggregates, and monomeric protein C. LF was fractionated in the same manner. The amount of aggregated protein (A + B) increased during mixing for both HF and LF (Figure 4a). In contrast, the amount of monomeric protein decreased (Figure 4b), suggesting that monomeric proteins might bind with other proteins (monomeric proteins or aggregated proteins) during mixing.

We investigated changes in the size distribution of protein aggregates in greater detail by evaluating the amounts of protein A and B aggregates (Figure 5). In HF, the amount of protein A aggregates decreased after the mixing peak was reached and that of protein B aggregates increased during mixing. These results indicate that the size of protein aggregates decreases during mixing. The amount of protein B aggregates could increase due to the binding of monomeric protein during mixing. The amount of protein B aggregates in LF increased during mixing, as in HF, but the amount of protein A aggregates did not significantly decrease. The amount of protein A aggregates increased up to the mixing peak, possibly due to the binding of monomeric proteins. 

### 3.3. Change in the Largest Protein Size Fraction during Mixing 

We used FFF to separate Protein Z into its component proteins. A typical FFF fractogram is shown in Figure 2b. In FFF, small proteins elute first and larger-sized proteins elute later. Protein Z aggregates was fractionated to give the largest proteins observed in this study. The amount of protein Z aggregates tended to decrease during mixing in HF (Figure 6) and more protein Z aggregates was present in LF than in HF. 

### 3.4. Change in the Strength of Non-Covalent Bonds during Mixing 

The 0.5% SDS buffer extracts contained proteins aggregated through the strongest non-covalent bond, the 0.3% SDS buffer extracts contained proteins aggregated through moderate non-covalent bonds, while the 0.1% SDS buffer extracts contained proteins aggregated through only very weak non-covalent bonds. We calculated the amounts of protein aggregated through non-covalent bonds of different strength as follows (Figure 7):

“The amount of protein aggregated through strong non-covalent bonds” = “the average amount (*n* = 6) of protein aggregates extracted with 0.5% SDS buffer” − “the average amount (*n* = 6) of protein aggregates extracted with 0.3% SDS buffer”; 

“The amount of protein aggregated through moderate non-covalent bonds” = “the average amount (*n* = 6) of protein aggregates extracted with 0.3% SDS buffer” − “the average amount (*n* = 6) of protein aggregates extracted with 0.1% SDS buffer”; and

“The amount of protein aggregated through weak non-covalent bonds” = “the average amount (*n* = 6) of protein aggregates extracted with 0.1% SDS buffer”. 

Differences in the amounts of protein extracted with SDS at various concentrations indicate the strength of non-covalent bonds, since SDS does not break disulfide bonds. Sonication at high power can break covalent bonds during protein extraction. In this study, however, sonication was conducted at a low power level which will not result in the breakage of disulfide bonds based on the original study [21].

In both HF and LF, the amount of protein aggregated through strong non-covalent bonds decreased during mixing, the amount of protein aggregated through moderate non-covalent bonds increased by the time of the mixing peak and then decreased, and the amount of protein aggregated through weak non-covalent bonds increased (Figure 8). These data indicate that non-covalent bonds between gluten protein aggregates weaken as dough mixing progresses. 

More proteins aggregated through strong non-covalent bonds in LF than in HF, indicating that non-covalent bonds in LF are stronger than in HF. Given the results shown in Figure 6, stronger non-covalent bonds in LF facilitate the formation of very large protein aggregates, resulting in long mixing peak times. 

### 3.5. Identification of Monomeric Protein Incorporated in Protein Aggregates during Mixing 

Electrophoretic images of the protein aggregates and monomeric protein in HF and LF are shown in Figure 9. Blue circles show protein spots whose intensities decreased during mixing (*n* = 4, *p* < 0.01, volume ratio > 2.0). Red circles show protein spots whose intensities increased during mixing (*n* = 4, *p* < 0.01, volume ratio > 2.0). The identification of each protein was estimated based on the report of Dupont et al. [26] and the identifications are shown in Table 1 and Table 2 (HF) and Table 3 and Table 4 (LF).

The amount of ω-gliadin in the monomeric protein in HF and LF decreased during mixing (Figure 9b,d, Table 2 and Table 4), indicating that the ω-gliadins were the most abundant monomeric protein incorporated into the protein aggregates during mixing. The amount of ω-gliadin in the protein aggregates tended to decrease (Figure 9a,c, Table 1 and Table 3), suggesting that once ω-gliadin is incorporated into a protein aggregates, there is further aggregation with other proteins and the aggregate becomes insoluble during mixing. In the monomeric protein fraction of HF, the amount of some enzymes and inhibitors (including serpins) decreased during mixing, possibly indicating that various enzymes and their substrates combine to form complexes during mixing.

In the monomeric protein fraction of HF, there was an increase in the low molecular weight (LMW)-glutenin subunit and α-gliadins, and in LF, the LMW-glutenin subunit and γ-gliadins increased during mixing (Figure 9b,d, Table 2 and Table 4). Omega-Gliadins were incorporated into the polymeric proteins during mixing, but some of these proteins disaggregated. In the protein aggregates in HF, the amount of high molecular weight (HMW)-glutenin subunit increased, and in LF, the amount of HMW-glutenin subunit and LMW-glutenin subunit increased during mixing (Figure 9a,c, Table 1 and Table 3), likely due to disaggregation of insoluble protein aggregates.

## 4. Discussion

Previous publications have shown that protein aggregates in dough became smaller during mixing [23,27,28] but it was difficult to interpret this behavior from size distribution results obtained using chromatography [25]. The amount of protein extracted from wheat flour is not the same as the amount of protein extracted from dough. Additionally, SE-HPLC has exclusion limits [22]. In this research, we used an extraction condition in which the amount of protein extracted from flour is the same as the amount of protein extracted from the dough. In addition to SE-HPLC analysis, we performed FFF analysis, which does not have a size exclusion limit, to analyze the size distribution of protein aggregates. The results of this study demonstrated that there are two major changes in gluten proteins during dough mixing: a decrease in the size of protein aggregates, and the binding of monomeric proteins (mostly ω-gliadins) to polymeric proteins.

This study investigated the mechanism underlying the decrease in the size of protein aggregates during mixing. Depolymerization of polymeric proteins [24], disaggregation [24,28], conformational rearrangement [29], and changes in protein surface properties [30] were previously identified as possible causes of increased protein solubility, and several reports describe the depolymerization of polymeric protein during dough mixing. Tanaka and Bushuk showed an increase in alcohol-soluble protein and acetic acid-soluble protein and a decrease in alcohol- and acid-insoluble residue in dough during mixing and attributed these changes to the depolymerization of high molecular weight glutenin [23]. MacRitchie [24] proposed that gluten proteins can be mechanically depolymerized by the cleavage of covalent bonds, based on Bueche’s theory [31]. Aussenac et al. performed SE-HPLC-MALS analysis of SDS un-extractable protein in dough and showed a decrease in un-extractable polymeric protein (UPP) due to the depolymerization of protein aggregates during mixing [32]. Many researchers have proposed the depolymerization of protein during mixing, but there were no data to reveal the relationship between non-covalent bonds and the size of protein aggregates. In this study, we demonstrated for the first time that weakening of non-covalent bonds is responsible for protein interactions during dough mixing. It appears that not only depolymerization but also disaggregation play major roles in decreasing the size of protein aggregates. Non-covalent bonds in proteins include ionic bonds, hydrogen bonds, and hydrophobic interactions. It is unclear which of these non-covalent bonds is most important, and thus further research is required.

We attempted to identify the monomeric proteins that bind to other monomeric and/or protein aggregates during mixing. 1D-PAGE is often used to identify protein components during dough mixing [33,34,35]. In this study we used 2D-DIGE, which has a higher resolution than 1D-PAGE and can accurately compare differences in the amount of protein in all spots. Previous studies have mostly focused on changes in the amount of HMW glutenin and LMW glutenin during dough mixing. Sievert et al. showed a decrease in HMW glutenin and LMW glutenin in residues during mixing by excluding acetic acid-soluble proteins, and concluded that the decrease in residue protein was responsible for the increase in the extractability of glutenin [33]. Skerritt et al. showed a decrease in HMW-glutenin subunit and B-LMW-glutenin subunuit and an increase in C-LMW-glutenin subunut and D-LMW-glutenin subunit in glutenin macropolymer (1.5% SDS insoluble protein) during mixing [34]. The authors claimed that disulfide bonds were cleaved and the polypeptides bound to free sulfhydryls in gluten proteins. In another report, Skerritt et al. showed that the amount of x-HMW-glutenin subunit in the glutenin macropolymer increased and the y-subunit decreased [35]. Aussenac et al. showed that the HMW-glutenin subunit in the glutenin macropolymer increased up to the mixing peak and then decreased, suggesting a relationship with dough breakdown [32]. Our 2D-DIGE data on HMW glutenin and LMW glutenin support the hypothesis of Sievert et al. In addition, our results show that ω-gliadins aggregated with other proteins and became insoluble. Since ω-gliadins do not contain cysteines, they must aggregate through non-covalent bond.

Fraction A + B and C are defined and discussed as “polymeric protein” and “monomeric protein” in the previous reports [12,20,36]. However, a small amount of ω-gliadins appeared to be present in this polymeric protein fraction. Since ω-gliadins do not have Cysteines and cannot form a disulfide bonds, they are most likely bounded to the polymeric glutenin proteins through strong non-covalent bonds. Fraction A + B consists largely of polymeric proteins of glutenin subunits linked through disulfide bonds. We think the polymeric proteins are aggregated each other and with monomeric proteins by non-covalent bonds in Fraction A + B, because we found that the higher the concentration of SDS was, the higher the quantity of the proteins in A + B was. So, we called the fraction A + B “protein aggregates” in this study.

This study showed the importance of non-covalent bonds in dough formation. The results shed more lights on our understanding of the mechanism underlying dough development. The preferential aggregation of ω-gliadins into large gluten polymers indicates the importance of non-covalent bonds in the formation of the gluten network in dough development.

## Figures and Tables

**Figure 1 foods-09-01643-f001:**
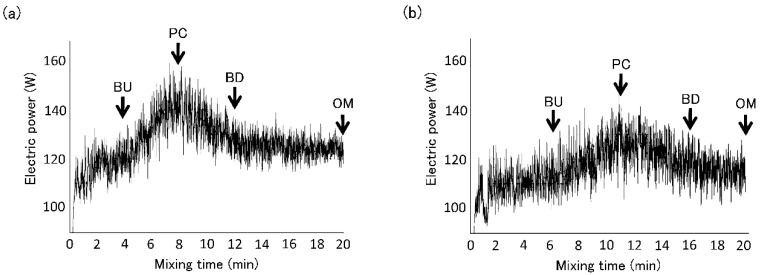
Dough mixing curves using high protein flour (**a**, HF) and low protein flour (**b**, LF). Arrows indicate sampling points. BU, PC, BD, and OM indicate build up, peak consistency, breakdown, and overmixing, respectively.

**Figure 2 foods-09-01643-f002:**
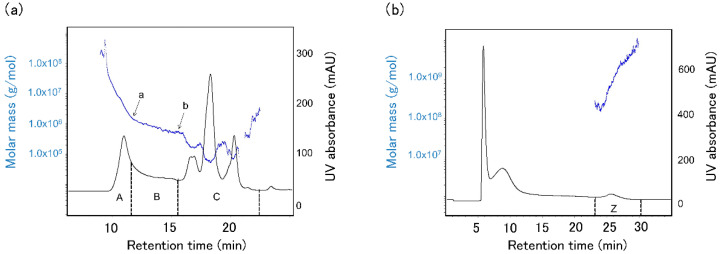
Size-exclusion high-performance liquid chromatography (SE-HPLC)-chromatogram (**a**, black), field flow fractionation (FFF)-fractogram (**b**, black) and molecular weight distribution curve (both, blue) obtained by MALLS analysis of protein extracted with 0.5% SDS buffer (pH 7.0) from high protein flour (HF). ‘a’ and ‘b’ are inflection points. ‘A’ ‘B’ is protein aggregates. ‘C’ is monomeric protein. ‘Z’ is defined as the largest-sized protein aggregates. Low protein flour (LF) was fractionated in the same manner.

**Figure 3 foods-09-01643-f003:**
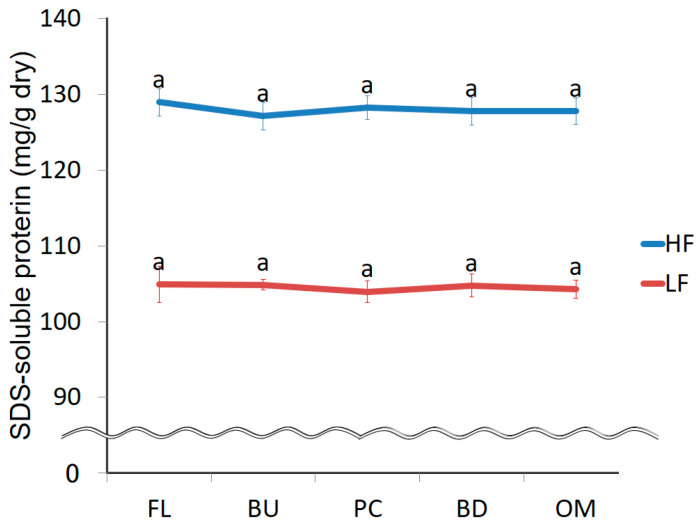
Change in the amount of 0.5% SDS-soluble protein during mixing. FL, BU, PC, BD, and OM indicate flour (before mixing), build up, peak consistency, breakdown, and overmixing, respectively. HF and LF mean high protein flour and low protein flour. The error bars show standard deviations *(n* = 12). Identical letters on the graph indicate no significant difference (*p* < 0.05).

**Figure 4 foods-09-01643-f004:**
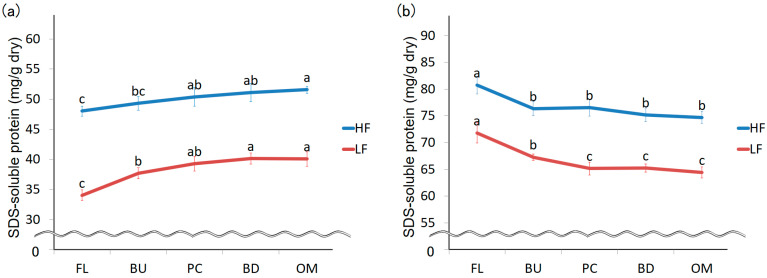
Changes in protein aggregates (**a**) and monomeric protein (**b**) extracted with 0.5% SDS buffer during mixing. FL, BU, PC, BD, and OM indicate flour (before mixing), build up, peak consistency, breakdown and overmixing, respectively. HF and LF mean high protein flour and low protein flour. The error bars show standard deviations (*n* = 6). Different letters on the graphs indicate significant differences (*p* < 0.05).

**Figure 5 foods-09-01643-f005:**
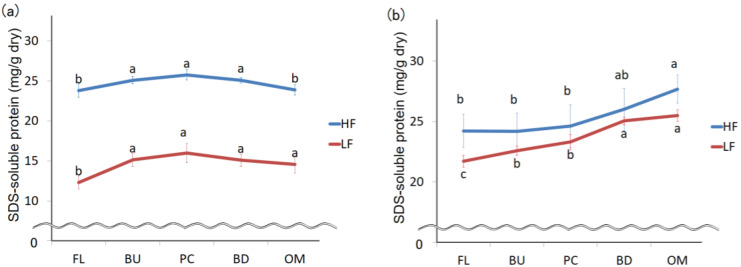
Changes in the amount of protein A aggregates (**a**) and porotein B aggregates (**b**) extracted with 0.5% SDS during mixing. FL, BU, PC, BD and OM indicate flour (before mixing), build up, peak consistency, breakdown and overmixing, respectively. HF and LF mean high protein flour and low protein flour. The error bars show standard deviations (*n* = 6). Different letters on the graphs indicate significant differences (*p* < 0.05).

**Figure 6 foods-09-01643-f006:**
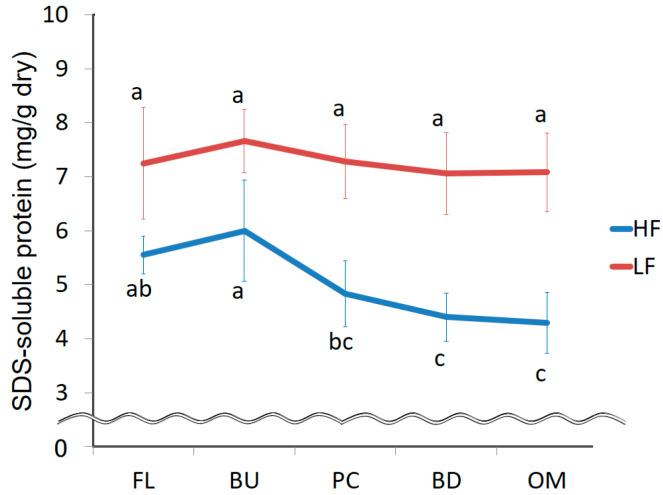
Changes in protein Z aggregates extracted with 0.5% SDS during mixing. FL, BU, PC, BD, and OM indicate flour (before mixing), build up, peak consistency, breakdown, and overmixing, respectively. HF and LF mean high protein flour and low protein flour. The error bars show standard deviations (*n* = 6). Different letters on the graph indicate significant differences (*p* < 0.05).

**Figure 7 foods-09-01643-f007:**
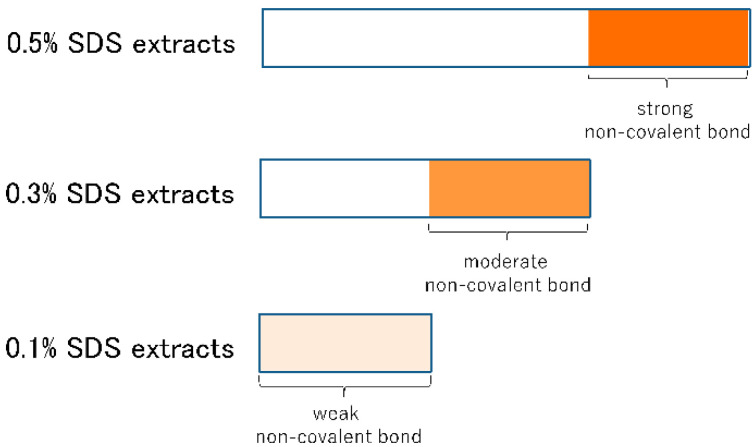
A descriptive sketch of the amount of protein aggregated through various strengths of non-covalent bonds. The amount of protein aggregated through strong non-covalent bond = “the average amount (*n* = 6) of protein aggregates extracted with 0.5% SDS buffer” − “the average amount (*n* = 6) of protein aggregates extracted with 0.3% SDS buffer”, moderate non-covalent bond = “the average amount (*n* = 6) of protein aggregates extracted with 0.3% SDS buffer” − “the average amount (*n* = 6) of protein aggregates extracted with 0.1% SDS buffer”, weak non-covalent bond = “the average amount (*n* = 6) of protein aggregates extracted with 0.1% SDS buffer”.

**Figure 8 foods-09-01643-f008:**
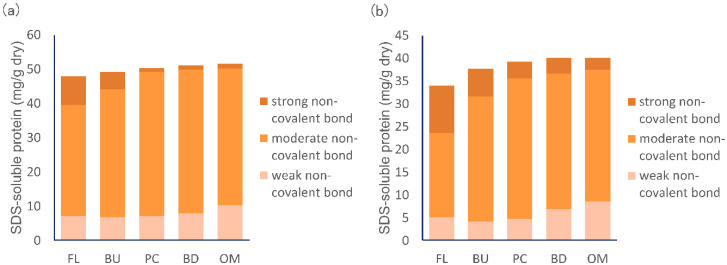
Changes in the amount of protein aggregated through various strengths of non-covalent bonds in high protein flour (**a**, HF) and in low protein flour (**b**, LF) during mixing. FL, BU, PC, BD, and OM indicate flour (before mixing), build up, peak consistency, breakdown, and overmixing, respectively. The amount of protein aggregated through strong non-covalent bonds = “the average (*n* = 6) of protein aggregates extracted with 0.5% SDS buffer” − “the average amount (*n* = 6) of protein aggregates extracted with 0.3% SDS buffer”, moderate non-covalent bond = “the average amount (*n* = 6) of protein aggregates extracted with 0.3% SDS buffer” − “the average amount (*n* = 6) of protein aggregates extracted with 0.1% SDS buffer”, weak non-covalent bond = “the average amount (*n* = 6) of protein aggregates extracted with 0.1% SDS buffer”.

**Figure 9 foods-09-01643-f009:**
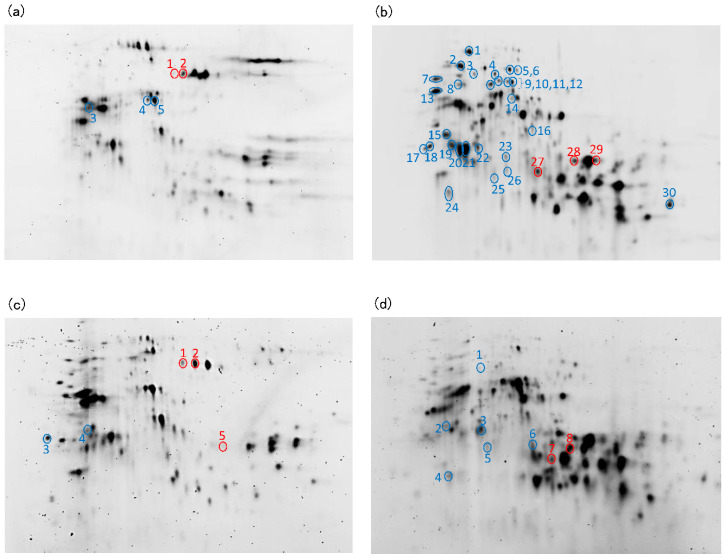
Electrophoretic images of the protein aggregates (**a**,**c**) and monomeric protein (**b**,**d**) in high protein (**a**,**b**, HF) and low protein flour (**c**,**d**, LF). Blue circles show protein spots whose intensities decreased during mixing (*n* = 4. *p* < 0.01, volume ratio > 2.0). Red circles show protein spots whose intensities increased during mixing (*n* = 4, *p* < 0.01, volume ratio > 2.0). Each spot number corresponds to the spot number in Table 1, Table 2, Table 3 and Table 4.

**Table 1 foods-09-01643-t001:** Decreased and increased proteins in protein aggregates in HF *^a^* during mixing.

Spot No. *^b^*	Identification *^c^*	Spot Ratio (%) *^d^*					Increase (↑) or Decrease (↓)
		Flour	Build up	Peak Consistency	Break Down	Overmixing	
1	HMW-glutenin	0.52 ± 0.25	0.64 ± 0.07	1.04 ± 0.09	1.12 ± 0.27	1.19 ± 0.17	↑
2	HMW-glutenin	0.59 ± 0.15	0.75 ± 0.05	1.17 ± 0.22	1.32 ± 0.27	1.36 ± 0.13	↑
3	ω-gliadin	1.12 ± 0.55	1.72 ± 0.54	0.74 ± 0.1	0.72 ± 0.24	0.85 ± 0.25	↓
4	ω-gliadin	1.66 ± 0.19	1.24 ± 0.18	0.73 ± 0.62	0.66 ± 0.38	0.56 ± 0.14	↓
5	ω-gliadin	2.08 ± 1.13	1.14 ± 0.34	0.69 ± 0.15	0.73 ± 0.13	0.62 ± 0.16	↓

*^a^* Spots were detected by 2dimensional fluorescence difference gel electrophoresis (2D-DIGE), significant differences (*p* < 0.05, *n* = 4) were determined by the ANOVA test. Proteins with volume ratios >2.0 are described. *^b^* These spot numbers correspond to the spot numbers in Figure 9a. *^c^* The identification of each protein was estimated based on the report of Dupont et al. [26]. *^d^* Values are means ± standard deviation.

**Table 2 foods-09-01643-t002:** Decreased and increased proteins in monomeric protein in HF *^a^* during mixing.

Spot No. *^b^*	Identification *^c^*	Spot Ratio (%) *^d^*					Increase (↑) or Decrease (↓)
		Flour	Build up	Peak Consistency	Break Down	Overmixing	
1	Enzymes	1.82 ± 0.39	0.51 ± 0.19	0.56 ± 0.15	0.46 ± 0.45	0.95 ± 0.91	↓
2	Enzymes	1.88 ± 0.35	0.54 ± 0.22	0.65 ± 0.21	0.42 ± 0.42	0.8 ± 0.38	↓
3	Enzymes	1.54 ± 0.87	0.44 ± 0.1	0.56 ± 0.11	0.46 ± 0.3	0.63 ± 0.27	↓
4	Enzymes	2 ± 1.22	0.57 ± 0.52	0.71 ± 0.38	0.45 ± 0.11	0.55 ± 0.13	↓
5	Enzymes	1.83 ± 0.88	0.39 ± 0.16	0.59 ± 0.16	0.4 ± 0.28	0.58 ± 0.53	↓
6	Enzymes	1.76 ± 0.83	0.23 ± 0.22	0.54 ± 0.26	0.41 ± 0.12	0.5 ± 0.29	↓
7	Enzymes	2.04 ± 0.79	0.56 ± 0.56	0.64 ± 0.29	0.35 ± 0.2	0.42 ± 0.34	↓
8	Enzymes	1.66 ± 0.42	0.74 ± 0.36	0.74 ± 0.19	0.46 ± 0.12	0.66 ± 0.36	↓
9	Enzymes	2.05 ± 0.98	0.53 ± 0.21	0.61 ± 0.21	0.52 ± 0.3	0.61 ± 0.3	↓
10	Enzymes	1.95 ± 1.19	0.67 ± 0.38	0.8 ± 0.39	0.54 ± 0.07	0.65 ± 0.23	↓
11	Enzymes	1.89 ± 1.1	0.48 ± 0.04	0.48 ± 0.18	0.33 ± 0.22	0.41 ± 0.37	↓
12	Enzymes	1.87 ± 0.7	0.5 ± 0.17	0.45 ± 0.25	0.37 ± 0.25	0.58 ± 0.77	↓
13	Enzymes	2.53 ± 1.07	0.64 ± 0.35	0.54 ± 0.27	0.35 ± 0.21	0.45 ± 0.45	↓
14	Enzymes	1.21 ± 0.32	0.76 ± 0.21	1.01 ± 0.34	0.49 ± 0.25	0.38 ± 0.18	↓
15	ω-gliadin	1.77 ± 0.73	0.93 ± 1.07	0.71 ± 0.16	0.41 ± 0.12	0.6 ± 0.28	↓
16	ω-gliadin	1.56 ± 0.33	0.82 ± 0.25	0.86 ± 0.04	0.71 ± 0.24	0.73 ± 0.3	↓
17	ω-gliadin	1.76 ± 0.34	0.65 ± 0.07	0.73 ± 0.11	0.62 ± 0.16	0.76 ± 0.26	↓
18	ω-gliadin	1.9 ± 0.37	0.78 ± 0.17	0.86 ± 0.13	0.62 ± 0.11	0.9 ± 0.33	↓
19	ω-gliadin	1.56 ± 0.47	0.44 ± 0.17	0.76 ± 0.23	0.52 ± 0.22	0.53 ± 0.29	↓
20	ω-gliadin	2.39 ± 1.15	0.7 ± 0.26	0.78 ± 0.17	0.51 ± 0.2	0.91 ± 0.16	↓
21	ω-gliadin	2.85 ± 0.99	0.63 ± 0.64	0.6 ± 0.2	0.38 ± 0.25	0.88 ± 0.76	↓
22	ω-gliadin	1.64 ± 0.35	0.6 ± 0.15	0.64 ± 0.17	0.75 ± 0.22	0.75 ± 0.34	↓
23	ω-gliadin	1.72 ± 1.06	0.47 ± 0.08	0.68 ± 0.28	0.51 ± 0.1	0.64 ± 0.41	↓
24	Serpins	2.28 ± 0.68	0.59 ± 0.15	0.8 ± 0.3	0.55 ± 0.17	0.53 ± 0.41	↓
25	Serpins	1.64 ± 0.59	0.74 ± 0.15	0.88 ± 0.15	0.68 ± 0.08	0.82 ± 0.24	↓
26	Serpins	1.7 ± 0.81	0.51 ± 0.09	0.79 ± 0.39	0.67 ± 0.12	0.71 ± 0.14	↓
27	α-gliadin	0.68 ± 0.19	1.47 ± 0.21	0.97 ± 0.16	1.18 ± 0.19	1.42 ± 0.65	↑
28	LMW-glutenin	0.78 ± 0.12	1.16 ± 0.09	1.16 ± 0.26	1.57 ± 0.52	1.41 ± 0.4	↑
29	LMW-glutenin	0.8 ± 0.11	1.21 ± 0.24	1.17 ± 0.1	1.63 ± 0.81	1.49 ± 0.3	↑
30	Inhibitors	1.32 ± 0.27	0.56 ± 0.25	0.54 ± 0.25	0.6 ± 0.16	0.51 ± 0.02	↓

*^a^* Spots were detected by 2D-DIGE, significant differences (*p* < 0.05, *n* = 4) were determined by the ANOVA test. Proteins with volume ratios > 2.0 are described. *^b^* These spot numbers correspond to the spot numbers in Figure 9b. *^c^* The identification of each protein was estimated based on the report of Dupont et al. [26]. *^d^* Values are means ± standard deviation.

**Table 3 foods-09-01643-t003:** Decreased and increased proteins in protein aggregates in LF *^a^* during mixing.

Spot No. *^b^*	Identification *^c^*	Spot Ratio (%) *^d^*					Increase (↑) or Decrease (↓)
		Flour	Build up	Peak Consistency	Break Down	Overmixing	
1	HMW-glutenin	0.7 ± 0.15	0.89 ± 0.1	1.51 ± 0.38	1.64 ± 0.17	1.8 ± 0.74	↑
2	HMW-glutenin	0.76 ± 0.19	1.09 ± 0.21	1.57 ± 0.45	1.47 ± 0.11	1.48 ± 0.64	↑
3	ω-gliadin	0.7 ± 0.16	0.81 ± 0.23	1 ± 0.37	1.94 ± 0.71	2.2 ± 1.2	↓
4	ω-gliadin	1.6 ± 0.37	1.35 ± 0.27	0.73 ± 0.07	1.11 ± 0.37	1.1 ± 0.2	↓
5	LMW-glutenin	0.7 ± 0.21	1.13 ± 0.21	1.6 ± 0.64	1.33 ± 0.21	1.3 ± 0.52	↑

*^a^* Spots were detected by 2D-DIGE, significant differences (*p* < 0.05, *n* = 4) were determined by the ANOVA test. Proteins with volume ratios > 2.0 are described. *^b^* These spot numbers correspond to the spot numbers in Figure 9c. *^c^* The identification of each protein was estimated based on the report of Dupont et al. [26]. *^d^* Values are means ± standard deviation.

**Table 4 foods-09-01643-t004:** Decreased or increased proteins in monomeric protein in LF *^a^* during mixing.

Spot No. *^b^*	Identification^c^	Spot Ratio (%) *^d^*					Increase (↑) or Decrease (↓)
		Flour	Build up	Peak Consistency	Break Down	Overmixing	
1	Enzymes	1.52 ± 0.58	0.48 ± 0.08	0.47 ± 0.2	0.35 ± 0.24	0.34 ± 0.27	↓
2	ω-gliadin	2.16 ± 0.8	0.88 ± 0.55	0.67 ± 0.1	0.55 ± 0.19	0.45 ± 0.24	↓
3	ω-gliadin	1.33 ± 0.16	0.87 ± 0.16	0.85 ± 0.07	0.88 ± 0.13	0.91 ± 0.1	↓
4	Serpins	2.48 ± 0.52	0.6 ± 0.12	0.48 ± 0.09	0.43 ± 0.22	0.48 ± 0.21	↓
5	Serpins	1.6 ± 0.22	0.74 ± 0.13	0.82 ± 0.1	0.83 ± 0.18	0.88 ± 0.33	↓
6	Enzymes	2.07 ± 0.42	1.21 ± 0.15	1.09 ± 0.42	0.88 ± 0.22	0.99 ± 0.25	↓
7	α-gliadin	0.63 ± 0.14	0.88 ± 0.24	1.21 ± 0.29	1.34 ± 0.16	1.53 ± 0.22	↑
8	γ-gliadin	0.84 ± 0.16	1.28 ± 0.23	1.52 ± 0.28	1.41 ± 0.07	1.56 ± 0.41	↑

*^a^* Spots were detected by 2D-DIGE, significant differences (*p* < 0.05, *n* = 4) were determined by the ANOVA test. Proteins with volume ratios >2.0 are described. *^b^* These spot numbers correspond to the spot numbers in Figure 9d. *^c^* The identification of each protein was estimated based on the report of Dupont et al. [26]. *^d^* Values are means ± standard deviation.

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
