# Peer review of "Changes in Protein Non-Covalent Bonds and Aggregate Size during Dough Formation"

_foods, 2020, doi:10.3390/foods9111643_

Round 1
Reviewer 1 Report
It is very interesting for cereal science to understand the contribution of non-covalent bonds in the aggregation of gluten proteins, comparing low and high protein flours, and clarifying the amounts and sizes of protein aggregates and monomeric proteins.
This article has high quality and contains new information, the aim of the work is clearly established, the experimental well conducted, discussion and conclusions are well documented and scientifically coherent, despite some concerns:
- Introduction:
- Lines 62-69 – “The amount of protein aggregates can be extracted depends on SDS concentration. With the increase of the concentration of SDS, the degree of disruption for non-covalent bonds increased. The proteins aggregated through weak non-covalent bonds are extracted with low concentration (0.1%, in this study) of SDS. The difference between the amounts of protein extracted with medium (0.3%) and low concentrations of SDS are macro-polymers aggregated through medium strength non-covalent bonds. Gluten protein aggregated through strong non-covalent bonds are calculated by the difference of the amount of proteins extracted with high (0.5%) and medium concentrations of SDS.” Which are the references for this methodology?
- Material and methods:
- Line 82 – “…distilled water (HF: 128.8 mL, LF: 117.4 mL; mean water absorption for farinograph)…”. Please indicate values in water absorption (%).
- Figure 1 – How BU, PC, BD and OM were calculated? Please clarify it in the text.
- Lines 94-97 – I am wondering if 10,000 rpm for 5 min, used to homogenize samples with SDS, is not too much, considering each dough was mixed only at 120 rpm for 20 min. It doesn´t have an effect in protein aggregates formation/ rupture? Why not add it during mixing? Maybe different results would be obtained adding SDS during the mixing step.
- Results:
- Figures 3, 4, 5 and 6 – Same letter(s) is(are) used both for HF and LF, despite the differences in soluble protein values (Fig. 3). The same comment for Fig 4, 5 and 6.
- Lines 207-208 – “The sizes of the protein aggregates decreased after the mixing peak in LF”. I don´t agree because there are no significant differences in LF.
- Lines 230-239 – There is a reference to this methodology?
- Discussion:
- Lines 319-320 – “…the extraction process disrupts the structure of the native protein”. How it was overcome in your case? Please add a comment about this aspect.
Author Response
It is very interesting for cereal science to understand the contribution of non-covalent bonds in the aggregation of gluten proteins, comparing low and high protein flours, and clarifying the amounts and sizes of protein aggregates and monomeric proteins.
This article has high quality and contains new information, the aim of the work is clearly established, the experimental well conducted, discussion and conclusions are well documented and scientifically coherent, despite some concerns:
Response: Thank you for the review of this manuscript. We have addressed each of your comments below.
1.Introduction: 1.Lines 62-69 – “The amount of protein aggregates can be extracted depends on SDS concentration. With the increase of the concentration of SDS, the degree of disruption for non-covalent bonds increased. The proteins aggregated through weak non-covalent bonds are extracted with low concentration (0.1%, in this study) of SDS. The difference between the amounts of protein extracted with medium (0.3%) and low concentrations of SDS are macro-polymers aggregated through medium strength non-covalent bonds. Gluten protein aggregated through strong non-covalent bonds are calculated by the difference of the amount of proteins extracted with high (0.5%) and medium concentrations of SDS.” Which are the references for this methodology?
Response: This is our original method. This is the first report that non-covalent bond was evaluated by extraction of various concentration of SDS. We got the idea from the previous report* as evaluation of disulfide bond by various concentration of dithiothreitol.
*Marie-Helene Morel et al. Biomacromolecules (2002),3,488-497
2.Material and methods: 128.Line 82 – “…distilled water (HF: 128.8 mL, LF: 117.4 mL; mean water absorption for farinograph)…”. Please indicate values in water absorption (%).
Response: The water absorption of HF is 64.4% and LF is 58.7%.
129.Figure 1 – How BU, PC, BD and OM were calculated? Please clarify it in the text.
Response: We wrote it in the manuscript. Please see Line 89-90.
130.Lines 94-97 – I am wondering if 10,000 rpm for 5 min, used to homogenize samples with SDS, is not too much, considering each dough was mixed only at 120 rpm for 20 min. It doesn´t have an effect in protein aggregates formation/ rupture? Why not add it during mixing? Maybe different results would be obtained adding SDS during the mixing step.
Response: We wanted to extract almost all protein as gently as possible. In our preliminary studies, we found 0.5% SDS was the lowest concentration that we can extract almost protein (Line 174-178). If we reduce the rotation speed or shorten the extraction time, the extraction efficiency will decrease. From the result, we decided the extraction condition (0.5%SDS, 10000rpm, 5min) for all experiments in this report. We think 10,000 rpm for 5 min is enough for our experiments because the extraction efficiencies is 91-93%.
Why not add it during mixing? Maybe different results would be obtained adding SDS during the mixing step.
Response: As you say, we think different results would be obtained adding SDS during mixing step. But our aim of this study is that analyzing of the change of the non-covalent bond in normal mixing condition.
3.Results: 6.Figures 3, 4, 5 and 6 – Same letter(s) is(are) used both for HF and LF, despite the differences in soluble protein values (Fig. 3). The same comment for Fig 4, 5 and 6.
Response: These graphs showed the change of the protein content of some fractions
in five mixing stages in each flour (HF or LF). We wouldn’t like to compare the HF and the LF. We did the statistical significance test in HF (or in LF). Therefore, we used the same letters in the same figures. Is it wrong how to show?
7.Lines 207-208 – “The sizes of the protein aggregates decreased after the mixing peak in LF”. I don´t agree because there are no significant differences in LF.
Response: It's just like what you said. We deleted the sentence (Line 211-212).
8.Lines 230-239 – There is a reference to this methodology?
Response: This is our original method. See our answer 1 for your question 1.
4.Discussion: 1.Lines 319-320 – “…the extraction process disrupts the structure of the native protein”. How it was overcome in your case? Please add a comment about this aspect.
Response: I am sorry, this problem is not overcome. We deleted this sentence (Line 333)

Reviewer 2 Report
L 94, “Immediately after sampling, the dough sample (1 g) was homogenized with 20 mL of 50 mM…” The two doughs should have different moisture content. Would this cause different solid content of the two samples for further analysis?
Fig. 2 A, please indicate which curve is for molar mass, and which for UV abs?
Session 3.1/Fig. 3. HF showed a higher protein protein than the LW, is this related or consistent with the protein content in the original flour? Please describe in the text.
L232-239, “the average (n=6) of…”, should use “the average amount (n=6) of…”, revise similarly for other definitions,
Fig. 7 and 8 showed very interesting results. Please elaborate in the discussion on what do you mean by “strong, moderate, and wheat non-covalent bond”, can you relate hydrogen, ionic and hydrophobic interactions to these bonds of different strength?
Author Response
Thank you for the review of this manuscript. We have addressed each of your comments below
L 94, “Immediately after sampling, the dough sample (1 g) was homogenized with 20 mL of 50 mM…” The two doughs should have different moisture content. Would this cause different solid content of the two samples for further analysis?
Response: As you pointed out, there is a slight difference in water content between each dough from HF and one from LF, but we do not think that it will affect the results of further analysis. We think that the difference in water content between LF and HF is small compared to the amount of 20 ml buffer added to dough, so it does not affect the results of further analysis.
Fig. 2 A, please indicate which curve is for molar mass, and which for UV abs?
Response: We revised Figure2.
Session 3.1/Fig. 3. HF showed a higher protein protein than the LW, is this related or consistent with the protein content in the original flour? Please describe in the text.
Response: Yes, it is. We revised manuscript. Please see Line 177-178.
L232-239, “the average (n=6) of…”, should use “the average amount (n=6) of…”, revise similarly for other definitions,
Response: Thank you. We revised manuscript. Please see Line 237-252.
Fig. 7 and 8 showed very interesting results. Please elaborate in the discussion on what do you mean by “strong, moderate, and wheat non-covalent bond”, can you relate hydrogen, ionic and hydrophobic interactions to these bonds of different strength?
Response: Thank you very much. We revised manuscript. Now, we have planned further analysis for ionic bonds or hydrophobic interaction to clarify the kind of non-covalent bonds.

Reviewer 3 Report
The manuscript by Iwaki et al investigates the aggregation state of wheat proteins during dough mixing using a combination of size fractionation and 2D gel electrophoresis. The work provide some new data but there are also some problems that needs to be addressed before it can be published.
Major issues:
1. The manuscript needs proper language editing and the text is sometimes difficult to understand.
2. No control experiments under reducing conditions are included to investigate the role of disulfide bonds.
3. The assignment of the 2D gels is based on previous studies. I would like to see some experimental validation of the assignment, e.g. MS of selected spots.
4. How were the "aggregate" and "monomer" fractions from SEC defined? How do the authors know that there are only monomers in the indicated parts of the chromatogram? Is it possible to do control experiment with fully disaggregated samples containing "only monomers"?
5. The largest aggregates may be removed by the filtration step (0.45 um for SEC or 1.2 um for FFF). Are those not important for dough properties?
Minor issues:
6. Results should not be described in methods section.
7. Line 35: write out "disulfide bond"
8. Line 38: why would hydrophobic interactions by prolines be stronger than other hydrophobic interactions?
9. Lines 170-172: "In a preliminary experiment, 0.5% SDS buffer was adopted so the amount of protein extracted with above 0.5% SDS buffer did not differ from the amount of protein extracted with 0.5% SDS buffer." What does this mean?
10. It would be interesting to see the corresponding SEC chromatograms for Fig 4 and 5 (maybe as supporting information?)
11. Figure 8: the amount of proteins do not add up to those in Fig 1. Why? FIs it related to the filtration mentioned in point 5?
12. Are the changes illustrated in Fig 8 significant?
13. Is there a difference between "polymeric" and "aggregated" proteins?
Author Response
The manuscript by Iwaki et al investigates the aggregation state of wheat proteins during dough mixing using a combination of size fractionation and 2D gel electrophoresis. The work provide some new data but there are also some problems that needs to be addressed before it can be published.
Response: Thank you for the review of this manuscript. We have addressed each of your comments below
Major issues:
- The manuscript needs proper language editing and the text is sometimes difficult to understand.
Response: We have made major revisions to our manuscript for proper wording by a native speaker. Please see the manuscript.
- No control experiments under reducing conditions are included to investigate the role of disulfide bonds.
Response: All experiments in this report relate to changes in non-covalent bonds, not disulfide bonds.
- The assignment of the 2D gels is based on previous studies. I would like to see some experimental validation of the assignment, e.g. MS of selected spots.
Response: Some studies* have identified proteins based on previous studies, so I referred to them.
*William J.Hurkman et al. J. Cereal Sci, 2004, 40, 295-299.
Malgorazata Szulc et al. Cereal Chem, 2009, 86, 692-694
Sajad Ahmad Sofi et al. Cereal Chem, 2020, 97, 85-94
Mike Sissons et al. Cereal Chem, 2019, 96, 193-206
- la Gatta et al. J. Cereal Sci, 2017, 73, 76-83.
- Lee et al. Cereal Chem, 2002, 79, 654-661
- How were the "aggregate" and "monomer" fractions from SEC defined? How do the authors know that there are only monomers in the indicated parts of the chromatogram? Is it possible to do control experiment with fully disaggregated samples containing "only monomers"?
Response: Please see Figure I in support file. Under non-reducing conditions (c and d), a part of protein of fraction (A+B) were stacked in the end part of stacked gel and the other proteins looked smeared in the high molecular region because they existed as “polymers” and/or “aggregates”, whereas the proteins of fraction (C) exist as only “monomers” because stacked protein or smeared proteins were not observed.
- The largest aggregates may be removed by the filtration step (0.45 um for SEC or 1.2 um for FFF). Are those not important for dough properties?
Response: Please see Figureâ…¡ in support file. There was no difference in the amount of protein between in the sample before and after filtration. The largest aggregates may pass through the filter.
Minor issues:
- Results should not be described in methods section.
Response: Neither Fig.1 nor Fig.2 is result. Figure1 shows some sampling points for analysis by typical mixing curve. Figure2 shows some fractions for analysis by typical SE-HPLC chart and FFF chart.
- Line 35: write out "disulfide bond"
Response: We revised manuscript. Please see Line 36.
- Line 38: why would hydrophobic interactions by prolines be stronger than other hydrophobic interactions?
Response: We revised manuscript. Please see Line 38-39.
- Lines 170-172: "In a preliminary experiment, 0.5% SDS buffer was adopted so the amount of protein extracted with above 0.5% SDS buffer did not differ from the amount of protein extracted with 0.5% SDS buffer." What does this mean?
Response: We revised manuscript. Please see Line 174-177.
- It would be interesting to see the corresponding SEC chromatograms for Fig 4 and 5 (maybe as supporting information?)
Response: We prepared SEC chromatograms. Please see Figureâ…¢ in support file.
- Figure 8: the amount of proteins do not add up to those in Fig 1. Why? FIs it related to the filtration mentioned in point 5?
Response: Figure8 shows the amount of protein aggregates. The amount of proteins in Figure8 matches the amount of proteins in Figure 4a.
- Are the changes illustrated in Fig 8 significant?
Response: Because it is the result of subtraction, we could not evaluate the significant difference. However, each extraction was done 6 times and the same result was got from each experiment.
- Is there a difference between "polymeric" and "aggregated" proteins?
Response: The protein bound by covalent bond (disulfide bond) is written as "polymeric protein". The protein aggregated by non-covalent bond is written as "protein aggregates".

Round 2
Reviewer 3 Report
I am still not satisfied with the authors response to major issues 2 and 4.
The authors write that "All experiments in this report relate to changes in non-covalent bonds, not disulfide bonds." If that is the purpose, they need to remove all the contributions of covalent (disulfide) interactions. Now they cannot determine if the changes between samples (different raw materials or processing time) comes form non-covalent interactions or formation/breakage of covalent bonds. Sonication can, for instance, provide enough power to break covalent bonds.
The Figure I in the supporting material (added in response to issue 4) is very informative. It shows a substantial change in the SDS-PAGE results between reducing and non-reducing conditions for fractions A & B. However, in contrast to what the authors claim, there is a change also for fraction C, which means that those proteins are also 'polymeric' (or maybe oligomeric). Moreover, according to the response to issue 13, the authors' definition of aggregated proteins are that they are kept together by non-covalent bonds. Obviously, the fractions A & B from SE-HPLC are primarily 'polymeric' (and maybe aggregated as well). In the text they are, however, referred to as aggregated.
I also have one additional issue based on the supporting chromatograms:
The chromatograms in Figure III shows that there is big difference in how the 3 peaks in the C-region respond to processing. The main change is in the peak around 20 ml while the other two peaks do not change very much. Is it then reasonable to analyze the whole region as one fraction?
Author Response
Thank you for reading my reply. We have addressed each of your comments below
I am still not satisfied with the authors response to major issues 2 and 4.
The authors write that "All experiments in this report relate to changes in non-covalent bonds, not disulfide bonds." If that is the purpose, they need to remove all the contributions of covalent (disulfide) interactions.
Response: Most of previous research* on molecular basis of dough development focused on covalent bonds, and the importance of disulfide bonds was very well documented. Dough lost the unique viscoelastic properties once the disulfide bonds in gluten proteins are broken with reducing agent. While their contribution to dough formation are well recognized, the nature of non-covalent bonds in gluten proteins is much less understood. Since dough cannot be developed in the presence of reducing agent, the impact of non-covalent bonds can only be investigated when disulfide bonds kept intact in the dough system.
* Tsen et al. Cereal Chem, 1963, 40, 399-407.
Bloksnma et al. Cereal Chem, 1972, 49, 104-118.
Schroder et al. Cereal Chem, 1978, 55, 348-360.
Okada et al. Cereal Chem, 1987, 64, 428-434
Skerritt et al. Cereal Chem, 1999, 76, 402-409
Aussenac et al. Cereal Chem, 2001, 78, 39-45
Morel et al. Biomacromolecules, 2002, 3, 488-497
Now they cannot determine if the changes between samples (different raw materials or processing time) comes form non-covalent interactions or formation/breakage of covalent bonds. Sonication can, for instance, provide enough power to break covalent bonds.
Response: Differences in the amounts of protein extracted with SDS at various concentrations indicate the strength of non-covalent bonds, since SDS does not break disulfide bonds. Sonication at high power can break covalent bonds during protein extraction. In this study, however, sonication was conducted at a low power level which will not result in the breakage of disulfide bonds based on the original study**. Previous studies have demonstrated that sulfhydryl-disulfide interchanges occur during dough mixing. These interchanges can impact non-covalent bonds among gluten proteins in the dough system.
** Singh et al. Cereal Chem. 1990, 67, 150-161.
The Figure I in the supporting material (added in response to issue 4) is very informative. It shows a substantial change in the SDS-PAGE results between reducing and non-reducing conditions for fractions A & B. However, in contrast to what the authors claim, there is a change also for fraction C, which means that those proteins are also 'polymeric' (or maybe oligomeric). Moreover, according to the response to issue 13, the authors' definition of aggregated proteins are that they are kept together by non-covalent bonds. Obviously, the fractions A & B from SE-HPLC are primarily 'polymeric' (and maybe aggregated as well). In the text they are, however, referred to as aggregated.
Response: Fraction C did not show any high molecular glutenin subunits under reducing condition. This fact indicates that there was no polymeric proteins in this fraction. It is still possible, however, some small oligomeric proteins with low molecular weight glutenin subunits only may also be present in fraction C. The reason why the band pattern appears to be different between the reducing condition and the non-reducing condition in fraction C is as follows: the intramolecular disulfide bonds of the protein are broken under reducing condition, and the protein molecules stretch and move slower in SDS-PAGE.
Previously, we had performed diagonal electrophoresis of all SDS-soluble proteins in wheat flour (Figure IV in the support file). If oligomers and polymeric proteins were present, spots should appear in red dot triangle, but no clear spots appeared. On the other hand, spots slightly off the diagonal indicate proteins in which the intramolecular disulfide bond is broken and shifted to the polymer side.
Based on above evidences, we are confident to define fraction C as a monomeric proteins.
Fraction A + B is defined and discussed as "polymeric protein" in the previous reports***. However, a small amount of ω-gliadins appeared to be present in this polymeric protein fraction. Since ω-gliadins do not have Cysteines and cannot form a disulfide bonds, they are most likely bounded to the polymeric glutenin proteins through strong non-covalent bonds. Fraction A + B consists largely of polymeric proteins of glutenin subunits linked through disulfide bonds. We think the polymeric proteins are aggregated each other and with monomeric proteins by non-covalent bonds in Fraction A + B, because we found that the higher the concentration of SDS was, the higher the quantity of the proteins in A + B was. So, we named the Fractions A + B "protein aggregates".
***Kuktaite et al. J. Cereal Sci, 2004, 40, 31-39
Yanaka et al. Breeding Sci, 2007, 57, 243-248
Markus Schmid et al. Cereal Chem, 2016, 93, 536-542
- la Gatta et al. J. Cereal Sci, 2017, 73, 76-83
Shewry et al., Wheat 4th ed., AACC International Press, 2009, pp. 223-298.
I also have one additional issue based on the supporting chromatograms:
The chromatograms in Figure III shows that there is big difference in how the 3 peaks in the C-region respond to processing. The main change is in the peak around 20 ml while the other two peaks do not change very much. Is it then reasonable to analyze the whole region as one fraction?
Response: Protein separation in SE-HPLC is not clear-cut. If the proteins are eluted completely follow the molecular size order, the Molar mass curve should be a downward-sloping line. However, it was wavy in Fraction C (Figure 2 in the manuscript). Therefore, monomeric proteins can only be evaluated in one fraction (the polymeric proteins were correctly eluted in order of molecular size, we were able to compare their sizes). As a result, the monomeric proteins that decrease during mixing were ω-gliadins, which have a relatively higher molecular weight when compared with other gliadins. Thus, we believe that it was appropriate to evaluate the monomeric proteins with one fraction.
